# Liquid Phase Infiltration of Block Copolymers

**DOI:** 10.3390/polym14204317

**Published:** 2022-10-14

**Authors:** Irdi Murataj, Eleonora Cara, Nicoletta Baglieri, Candido Fabrizio Pirri, Natascia De Leo, Federico Ferrarese Lupi

**Affiliations:** 1Advanced Materials Metrology and Life Sciences Division, Istituto Nazionale Ricerca Metrologica (INRiM), Strada delle Cacce 91, 10135 Torino, Italy; 2Department of Applied Science and Technology, Politecnico di Torino, Corso Duca degli Abruzzi, 24, 10129 Torino, Italy

**Keywords:** block copolymers, BCPs, self-assembly, liquid phase infiltration, LPI, metal loading

## Abstract

Novel materials with defined composition and structures at the nanoscale are increasingly desired in several research fields spanning a wide range of applications. The development of new approaches of synthesis that provide such control is therefore required in order to relate the material properties to its functionalities. Self-assembling materials such as block copolymers (BCPs), in combination with liquid phase infiltration (LPI) processes, represent an ideal strategy for the synthesis of inorganic materials into even more complex and functional features. This review provides an overview of the mechanism involved in the LPI, outlining the role of the different polymer infiltration parameters on the resulting material properties. We report newly developed methodologies that extend the LPI to the realisation of multicomponent and 3D inorganic nanostructures. Finally, the recently reported implementation of LPI into different applications such as photonics, plasmonics and electronics are highlighted.

## 1. Introduction

The development of new strategies for the fabrication of nanostructured materials with tailored properties has been of intense interest among the scientific community in recent years. Novel nanomaterials with great control over their geometrical and functional features are the focus of research nowadays in the continuous search for device shrinkage with boosted performances. Although conventional lithographic techniques, i.e., optical and electron lithography, represent the main lithographic tools in the semiconductor industry, key challenges are emerging in the delivery of increasingly powerful and smaller devices due to the high processing cost and complexity. To this end, self-assembling materials, such as block copolymers (BCPs), are of potential use for nanopatterning thanks to their ability to provide a wide range of structures at the nanoscale down below the optical lithography resolution limit [1]. BCPs consist of two or more different and incompatible homopolymer chains linked together by a covalent bond. Under specific annealing conditions, the amphiphilic nature of the BCPs is responsible for microphase separation, that is, the so-called self-assembly. This generates the in-parallel self-registration of periodic structures at the nanoscale in the typical range of 10-100 nm. These materials offer a great tunability in terms of dimensions, morphologies and pattern orientation that can be manipulated in the synthetic phase by the molecular weight, volume fraction and substrate functionalisation [2,3]. So far, self-assembled BCPs have been exploited for several applications spanning nanolithography [4], electronics [5], photonics [6], energy [7,8] and membranes [9].

The ever-increasing versatility of BCPs is given by the presence of functional groups carried by their polymeric chains. The functional moieties act as reactive centres for the nucleation and growth of inorganic materials upon selective infiltration of specific precursors, providing ideal templates for the synthesis of hybrid organic-inorganic or all-inorganic materials [10]. Vapour phase infiltration techniques, such as sequential infiltration synthesis (SIS), offer exquisite control over the fabrication of nanostructured inorganic films [11]. A lot of effort has been dedicated to the extension of the library of materials that can be infiltrated by SIS, however, primarily leading to the synthesis of metal oxides [12] with a few exceptions of metals [13]. SIS, along with other vapour phase methodologies, i.e., vapour phase infiltration (VPI), micro-dose infiltration synthesis (MDIS) and multipulse vapour infiltration (MPI), shares the atomic layer deposition (ALD) system. The sequential exposure of self-assembled BCPs to specific metalorganic precursors and oxidising agents (H_2_O, H_2_O_2_, O_3_) determines the selective growth of metal oxides inside the constituent block, which carries the appropriate chemical functional group. Thus, after the selective removal of the organic matrix, the initial template is replicated [14]. However, due to the numerous processing variables involved, such as processing temperature, pressure, pre-treatments, precursor and oxidising agent exposure times and polymer–precursor reactivity, SIS suffers from a high complexity in the choice of the experimental design parameters as well as the requirement for expensive equipment and hazardous metalorganic precursors. On the other hand, liquid phase infiltration (LPI) processes have emerged as an overall much simpler fabrication avenue for the selective growth of several inorganic materials in self-assembled BCPs, requiring metal salt chemicals soluble in H_2_O or ethanol and standard lab equipment [15].

In the literature, many different terminologies have been used to describe this process, such as LPI [16,17], ion adsorption [18], aqueous metal reduction [19], metal ion loading [15], metal inclusion [20] and metal incorporation [21]. Although each one might indicate a different interaction mechanism between the metal precursor and the polymeric template (complexation or electrostatic) or different process conditions (neutral or acidic), they all can be referred to as a liquid phase route to selectively localise metal salt precursors into one of the BCP nanodomains. We encourage the use of the term LPI to describe the overall process, while other names, such as metal loading/inclusion/incorporation, might be better suited to specifically refer to the metal salt precursor localisation into one of the two BCP domains. A united and cohesive use of terminology in the literature would ensure an easier diffusion of the results within the scientific community.

Compared to SIS, LPI offers a wider range of metallic elements (Au, Pt, Pd, Fe, Co, Cu, Ni) [15,22,23], metal alloys (FePt, AuAg) [24,25,26], multicomponent nanopatterns (Pd/Pt, Au/Pt) [27], metal oxides (Fe_2_O_3_, Fe_3_O_4_, NiO, Cr_2_O_3_) [16,20,28] as well as metal halide perovskites (MAPbBr_3_) [21] that can be incorporated inside BCPs commonly used in nanopatterning applications such as poly(styrene)-*b*-poly(2-vinylpyridine) (PS-*b*-P2VP), poly(styrene)-*b*-poly(4-vinylpyridine) (PS-*b*-P4VP) and poly(styrene)-*b*-poly(ethylene oxide) (PS-*b*-PEO). Although LPI was initially exploited for lithographic purposes [29,30,31], recent developments in infiltration design have led to the synthesis of materials with added functionalities, widening its application to plasmonics [32] and electronics [33]. In this review, we report the latest advances in the LPI of inorganic materials, specifically focusing on BCPs as templates for the synthesis of novel functional materials. The infiltration mechanism and the material properties will be addressed to give insights into future applications.

## 2. LPI Processing and Mechanism

LPI of BCPs is a wet chemical process consisting of selective binding via complexation or electrostatic interaction of a metal salt precursor and a reactive functional group carried by one of the constituent blocks. The following polymer removal and simultaneous reduction or oxidation of the metal precursor, whether by O_2_ plasma, thermal degradation or UV/ozone treatment, reveals a nanostructured inorganic material (typically metals or metal oxides) whose morphology replicates that of the BCP template [34] (Figure 1). The selective incorporation of the precursor could be achieved either prior to the BCP self-assembly process, by mixing the BCP and metal salt solution [28], or afterwards, by soaking the self-assembled nanostructures into the precursor-containing solution [17]. Alternatively, the metal salt solution can be directly spin-coated on top of the polymeric film [32].

The distinct chemical functionalities of the constituent blocks of the BCP template are the key factors leading to the selective binding of the precursors. The seminal work by Chai et al. [34] reported the fabrication of continuous metallic nanowires using horizontal cylindrical PS-*b*-P2VP by soaking the self-assembled BCPs into acidic solutions of Au, Pd and Pt metal salts. The selective binding of the precursors in the P2VP block is related to the Brønsted base character of the pyridine moiety. In mild acidic environments, pyridine-containing polymers such as P2VP can be easily protonated and therefore they can bind negatively charged anionic metal complexes ([AuCl_4_]^−^, [PdCl_4_]^−^, [PtCl_4_]^−^) derived by the metal salt precursor, by electrostatic interaction (Figure 2a).

Due to the lack of any positively charged functional group in PS, the anionic complexes selectively locate inside the P2VP cylindrical domains only. The subsequent brief exposure to O_2_ plasma simultaneously removes the polymer template and reduces the metal salt to Au, Pd and Pt nanostructures that replicate the BCP morphology. In contrast to vertically aligned PS-*b*-P2VP cylinders, horizontally oriented cylinders do not have direct access to the metal salt precursor solution due to the PS hydrophobic matrix surrounding the P2VP nanodomains. Therefore, the acid concentration (i.e., HF and HCl) plays a key role in guaranteeing the formation of continuous metallic nanowires. The effect of acidic conditions on the LPI is related to the surface reconstruction of the BCP nanostructures. The repulsive interactions between the protonated pyridine groups lead to the selective swelling of the P2VP cylinders perforating the PS overlay, hence fully exposing the reactive nanodomains to the salt solution (Figure 2b). The surface reconstruction has been demonstrated to be critical also for the successful infiltration of vertically aligned PS-*b*-P4VP and PS-*b*-PEO cylinders. Although being the P4VP and PEO nanodomains in direct contact with the precursor solution, an ‘activation step’ by means of a preferential solvent (ethanol or butanol) induces a selective swelling of the polar domains. This process slowly modifies the surface without changing the structural arrangement and dimensions of the BCP but yields a nanoporous film that facilitates the sorption of metal cations (Fe^3+^, Cu^2+^, Ni^2+^, Cr^3+^) via coordination with the P4VP and PEO domains [16,35,36,37].

## 3. Metal Salt Diffusion and Reduction

The successful replication of BCP templates into high-quality inorganic materials depends on the precursor infiltration and conversion processes. The control over the different parameters involved in precursor diffusion, precursor-BCP selective binding and precursor reduction or oxidation is crucial to the design of materials with tailored compositions and added functionalities. For instance, as reported by Subramanian et al. [17], LPI at mildly elevated temperatures (40–80 °C) considerably improves the infiltration kinetics of the metal salt precursor into the BCP nanodomain. LPI at higher temperatures enhances up to six the molar ratio of infiltrated Pt per V2P monomeric unit for an overall increased metal uptake when compared to room temperature LPI. The control over the quantity of material infiltrated inside the BCP domains is reflected in a better quality of the inorganic pattern thanks to a higher fidelity to the starting BCP template, also enabling the tuning of the optical properties of the resulting material (Figure 3).

The control of the diffusion of metal salt precursors along the thickness of thick BCP templates is a key requirement for the achievement of high aspect-ratio or complex 3D metallic nanostructures. High aspect-ratio 3D nanoporous Pt structures with enhanced catalytic activity for hydrogen evolution reaction were fabricated by LPI of thick (120 nm) lamellar BCP templates with two pyridine-containing blocks as P2VP-*b*-P4VP. Although the constituent blocks P2VP and P4VP are both capable of interacting with metal salt precursors, P4VP shows a stronger counterion binding when compared to P2VP [38]. Due to the difference in the binding power of the two blocks, the metal salt precursors are preferentially allocated in the P4VP domains. Therefore, after the polymer removal and precursor reduction, the metal in the P4VP domains becomes continuous whereas the metal in the P2VP is not, resulting in a 3D nanoporous structure [39]. The high metal salt precursor diffusion, along with proper acid concentrations, were recently exploited to reveal complex multilevel morphologies of thick (200 nm) cylinder-forming PS-*b*-P2VP, where higher acid concentration led to an increase in the cylinder diameter thanks to the larger uptake of metal ions [23] (Figure 4).

Fine tuning of the content of infiltrated precursors reveals to be crucial in the definition of the metallic nanostructure shapes with a direct impact on the optical properties of the resulting material. Alvarez-Fernandez et al. [32] showed that one can fabricate high-refractive index surfaces by varying the immersion duration of PS-*b*-P2VP into the metal precursor solution ([AuCl_4_]^−^). The morphology of the nanofeatures can be tuned depending on the effective quantity of infiltrated precursors into the polymeric template. The subsequent exposure to O_2_ plasma reveals the growth of Au as individualised dots for short immersion times (1 h in Figure 5A) or rod-like particles of varying lengths for longer immersion times (48 h and 120 h in Figure 5B,C, respectively).

## 4. Multicomponent Materials and Complex Structures

As mentioned in the introduction paragraph, the search for novel materials with added functionalities has driven the research towards the extension of LPI to a wide number of metals and to the development of sophisticated synthetic routes for the infiltration of more complex materials. LPI of BCPs allows for great control over the nucleation and growth of different multicomponent materials in which the structural/compositional distribution can be arranged by the simultaneous or sequential infiltration of different metal salt precursors. A paradigmatic example is represented by the fabrication of segmented multicomponent nanowires by exploiting the reversible complexation of metal ions with protonated P2VP of PS-*b*-P2VP in acidic conditions, as reported by Mun et al. [40]. Here, after a first infiltration of [PtCl4]− into the P2VP domains, the immersion of the infiltrated polymer into a concentrated acidic solution allowed the reversal of the complexation of metal anions, leaving the P2VP available for a subsequent infiltration with another metal anion complex as [AuCl4]−. The authors extended this approach by further implementing optical lithography on top of the self-assembled nanostructures. The optical mask drives the selective replacement of a metal anion in a specific spatial location dictated by the optical pattern. The final O_2_ plasma treatment reveals continuous metallic nanowires with different metallic components. The same authors unveiled a generalised route to synthesise intermetallic nanoalloy arrays using cylindrical PS-*b*-P2VP as a starting template [24]. Various bimetallic (FePt, CoPt, PdAu) and trimetallic (CoPdPt) alloys were obtained by the simultaneous loading of multiple ionic metal precursors on the self-assembled BCPs, offering precise control over size, composition and single-crystalline intermetallic atomic structures of the nanoalloys (Figure 6a).

On the contrary, for the synthesis of alloys such as AuAg, a specific order in which the metal salt precursors (HAuCl_4_ and AgNO_3_) are added is required to avoid undesired precipitations of highly insoluble AgCl [26]. Core-shell Au/Ag nanoparticles and composites, however, can be synthesised by exploiting the different chemical affinities of the P2VP and PEO blocks towards HAuCl_4_ and AgNO_3_ in micellar triblock PS-*b*-P2VP-*b*-PEO or in PS-*b*-P2VP/PS-*b*-PEO blends. The positively charged P2VP in acidic conditions preferentially interacts with Au salt anions while repelling Ag+ ions that are mainly localised inside the PEO nanodomain (Figure 6b) [25,35]. The ongoing research extended the BCP liquid phase infiltration to the synthesis of new classes of materials such as metal halide perovskites [21]. In the mixture of MABr (methylammonium bromide) and PbBr2 and PS-*b*-P2VP, the precursor ions preferentially coordinate with pyridine moieties in the P2VP blocks via Lewis acid–base interactions. Nanoimprinting lithography by means of a pre-patterned PDMS mould is used to simultaneously crystallise methylammonium lead bromide (MAPbBr_3_) perovskite and microphase separate the loaded PS-*b*-P2VP leading to a hierarchically ordered MAPbBr_3_ embedded in the BCP template.

The great versatility of LPI for the realisation of complex nanoarchitectures is demonstrated by its successful integration with the most recent BCP nanopatterning techniques [33,41,42,43,44]. The nanodomain swelling of the hydrophilic BCP domains is a general route to the morphology tuning of the resulting inorganic nanostructures. As an example, starting from cylindrical PS-*b*-P2VP, a different morphology evolution can be induced on the BCP nanopattern during the solvent annealing, the metal loading and polymer removal step, generating metallic nanopatterns with diverse shapes such as hexagonal nanorings, hexagonal nanomesh and double lines [22]. Shin et al. [27] revealed a great example of elaborate nanopatterns integrating different metallic components, whose lateral ordering can be precisely controlled by directed self-assembly (DSA). After a first DSA of PS-*b*-P4VP by means of graphoepitaxy and subsequent LPI, a second level of BCP was applied on top of the resulting metallic nanofeatures to create a superimposed nanopattern. Depending on the morphology of the two BCP layers (cylinders or micelles) and on the different metal salt precursors used at distinct stages, one can tailor the fabrication of multilevel nanopatterns with complex structures and diverse chemical compositions such as Pt nanowire-Pd nanodot arrays or Au-Pt nanodot arrays (Figure 7a).

A similar multilevel approach was used by Subramanian et al. [17] for the fabrication of Pt nanomeshes. Upon sequential deposition of lamellar BCP layers and subsequent Pt infiltration, the authors were able to stack lamellar Pt nanowires oriented orthogonally to the lamellar features underneath, resulting in a 3D Pt nanomesh structure in which the electrical conductance is dependent on the number of stacked layers (Figure 7b). A further extension of the same multilayer fabrication route is represented by its integration into topographical patterns [43]. In this case, the authors take advantage of different mechanisms involved in the alignment process while performing DSA. In this multimechanism directed self-assembly (MMDSA) approach, the first BCP layer alignment with respect to the guiding pattern is dictated by a proper choice of the graphoepitaxy parameters in terms of trench depth and BCP film thickness, whereas the second BCP layer alignment is directed orthogonally to the underlying BCP layer. By finely tuning the alignment mechanism and the LPI processes involved in each step, one can control the orientation of single component (Pt) or multicomponent (Pt-Fe) nanomeshes with respect to the substrate guiding patterns (Figure 7c). In the multilayer stacking approach, although being very valuable for the fabrication of 3D metallic or multicomponent nanomeshes, the spontaneous orthogonal alignment of BCP nanofeatures with respect to those of the lower level limits the library of achievable structure symmetries. Photothermal techniques such as laser-zone annealing (LZA), in combination with the soft-shear (SS) effect of an elastic polymer cladding, enable the formation of uniaxially aligned BCPs depending on the direction of the focused laser line sweep across the film. Majewski et al. [33] demonstrated that when the BCP nanostructures are converted into metallic replicas by LPI, one can deposit a subsequent layer of BCP and perform SS-LZA to align each layer independently in any desired symmetry based on the relative laser sweep direction, leading to metallic nanomeshes with arbitrary cross-angles (Figure 8a). Independent control of the BCP self-assembly and spatial location of the resulting nanopatterns were achieved by Kim et al. [44] by combining shear-induced alignment with conventional optical lithography. First, the authors applied shear stress over the BCP film with a cured poly(dimethylsiloxane) (PDMS) pad and subsequently immersed the shear-aligned nanostructures into a metal salt precursor solution. Afterwards, a higher level of microstructures was created on top of the loaded BCP by a photolithography process. The subsequent metal salt precursor reduction by O_2_ reactive ion etching (RIE) on the developed areas followed by the lift-off process, reveals highly-ordered Au nanowires in well-defined micrometric areas dictated by the lithographic design (Figure 8b).

## 5. Applications

The inclusion of metallic components via the liquid phase infiltration process in block copolymers nanoscale templates has created many opportunities for the realisation of complex metallic nanostructures, thus expanding the applicability of BCPs lithography itself. LPI offers the possibility to accurately replicate the nanoscopic intricate templates of different BCPs morphologies overtaking traditional physical vapour deposition processes in the realisation of composite nanomaterials, where the lift-off step is posing limitations at sub-20 nm resolution. In this section, some recent works implementing LPI of BCPs in photonics, plasmonics and electronics are discussed as well as disparate applications in different domains of materials science.

### 5.1. Photonics and Plasmonics

In the realm of photonics and plasmon-enhanced molecular spectroscopies, one of the research focuses in the last decades has been the realisation of metallic nanostructures with controlled dimensional features and optical properties, e.g., localised surface plasmon resonances (LSPR) frequency and refractive index, to boost the applicability of these methods as well as to study fundamental phenomena not yet explored. The specific requirements, for standardisation, include reproducible methods and reliable and rationally designed substrates with tailored optical and plasmonic properties [45]. In this view, BCPs have been proven to offer excellent lithographic templates with ordered nanoscale morphologies for patterned surfaces extended over square centimetre areas [21].

The reproducibility of the self-assembly process with controllable and repeatable dimensionality bound to the polymers’ chemical properties (e.g., size and pitch related to molecular weight) guarantees reliable modelling of the optical near and far field properties of the metallic replicas. In this framework, the LPI of metals inside BCP templates has already been explored since the late nineties for the realisation of 1D nanoparticles, 2D planar systems and 3D structures with low aspect ratios. Some early applications of the process, sometimes referred to as in situ reduction of metal ions in polymeric domains, have been applied to the realisation of 1D plasmonic nanocomposites, i.e., metallic nanoparticles embedded in a dielectric matrix. BCPs micellar domains have been infiltrated to tune the critical dimension of the plasmonic nanocomposites down to 2 nm [46] and to regulate LSPR frequencies and refractive index [47]. Other 1D nanoparticles with bimetallic content of Au and Ag have also been characterised with UV-Vis absorption spectroscopy finding a linearly dependent plasmon resonance frequency, which could be tuned with the relative content of Au [26].

Pure Ag or Au nanoparticles presented an absorption peak at 415 nm or 538 nm, respectively, while in the varying range between Au:Ag = 1:3 and Au:Ag = 3:1, the absorption peak shifted from 430 nm to 515 nm, respectively. More recently, single or bi-metallic (Au/Ag) loading in BCPs micelles has been combined with their self-assembling tendency to produce a monolayer coverage on the surface of scanning probe microscopy (SPM) tips to support plasmonic enhancement and morphological characterisation in TERS-like configuration [48]. The results of Zito et al. are reported in Figure 9, highlighting (a) the bi-metal loading process in BCPs micelles, (b) the tip covered in plasmonic nanocomposites and (c) the computational results showing the enhanced electric field at the tip site.

Gold nanoparticles in octahedral, decahedral and icosahedral shapes have also been obtained in a single-step synthesis through PS-*b*-P2VP loading with HAuCl_4_, where the processing temperature and polymer-metal ratio have been deemed essential in the shape and size dispersity of the nanoparticles batch [49]. Such nanoparticles have also been tested as SERS-active platforms for different thermal treatments where the increasing temperature implies a reduced polymer shell around the nanoparticle, which is compatible with higher enhancement activity.

Other 1D materials with controllable optical properties have been reported by Glass and coworkers combining block copolymer micelle nanolithography (BCMN) with standard lithographic methods. The authors reported that Au nanodots can be used to epitaxially grow ZnO nanoposts, whose diameter might be tuned in such a way that they can be applied as sources of coherent light [50]. Two-dimensional composite structures have been obtained as multilayered nanostructures alternating a polymer lamella with one loaded with metal salts, assembled in parallel to the substrate [51]. The group of Kim et al. exploited the LPI of [AuCl4]− salt in P4VP cylindrical template and the swelling of BCPs as a tuning process for the metallic array morphology evolving to dot, ring or hexagonal meshes [22]. The UV-Vis transmittance spectra of the resulting planar structures were measured and simulated using the hexagons’ critical dimensions, and the electric field intensity profiles were simulated in the xy plane at z = 2 nm above the surface under resonance conditions at 550 nm. Another significant use of LPI was reported by Haridas and Basu, who realised a two-dimensional hybrid array of CdSe quantum dots (QDs) confined in P4VP nanodomains, which were further surrounded by plasmonic gold nanoparticles (AuNPs) obtained through chemical reduction of gold chloride in thiol-terminated polystyrene [52]. The photoluminescence of the hybrid array was dependent on QDs-AuNPs coupling through the overlapping of the spectral features of the exciton and the plasmon bands. Other two-dimensional structures were designed by Shin and coworkers, who combined multicomponent nanopatterns with double-metal infiltration to obtain Au nanowires acting as optical antennas alternated with Pt nanodots creating the conditions for localised hot spots or “photon sinks” [27]. Recently, Ponsinet and coworkers developed a high refractive index plasmonic surface obtained by gold salt reduction inside PS-*b*-P2VP out-of-plane lamellae [32].

The plasmon resonance frequency and refractive index were extracted by spectroscopic ellipsometry (SE), finding a shift in the SPR spectral position from 539 nm to 563 nm with increasing gold content obtained after 1, 48 and 120 h of LPI process time, already reported in Figure 5. The SE was also analysed to extract the optical constants reported in Figure 10a, yielding a refractive index of 3.2 corresponding to the maximum infiltration time and to a low metal fraction with respect to literature values. The same group developed 3D arrays of hybrid Al2O3/Au raspberry-like nanoclusters obtained by vapour-phase infiltration of P2VP nanodomains in PS-*b*-P2VP out-of-plane lamellar template, and Au decoration via metallic salt reduction in a second BCPs template [42], as schematised in Figure 10b. The nanostructures were characterised by AFM and Fourier fast transform analysis, highlighting the presence of two different periodicities corresponding to the first BCPs layer infiltrated via SIS with Al2O3 (referred to as process step C and D in Figure 10b) and the second self-assembled layer impregnated with Au (step F in Figure 10b). These 3D composite nanostructures are envisioned to be suitable for the realisation of optical metamaterials and label-free sensing platforms.

### 5.2. Electronics

In the domains of micro and nanoelectronics, the high patterning tunability offered by BCP is making the nanostructures effectively applicable for their integration with existing semiconductor technologies, with large versatility under the user’s control. Once the sub-20 nm scale challenge has been overcome, the research is focused on the exploitation of the enhanced properties of the assembled nanostructures with respect to the isolated ones. Recently, Liu et al. [43] obtained Pt nanowires through MMDSA (Figure 7c) showing electrical continuity, thus raising interest in interconnects. As reported in Figure 11a, the measured resistance corresponds to a resistivity on the order of 10 Ωcm, that is, orders of magnitude larger than that of bulk Pt (10−5 Ωcm).

Indeed, the enhanced scattering of conduction electrons at the wire surfaces enlarges the resistivity, while stacking the wires into meshes could be a good strategy to increase the conductivity. A similar result, in terms of electrical resistivity and ohmic conduction, was reported by Chai and coworkers [34] concerning aligned platinum nanostructures showing clear conductivity over a length scale up to micrometres. More precisely, they noted that increasing the Na2PtCl4 concentration from 0.1 mM to 1 mM and 10 mM in the LPI solution produced wires with larger diameters (from ~7 nm to ~12 nm) and, correspondingly, a resistance per unit length varying from 3∙106 Ω/µm to 1.2∙106 Ω/µm. Analogously, as shown in the scatter plot in Figure 7b, Subramanian et al. [17] tested platinum nanomesh structures where a single lamellar pattern has no measurable conductance, which instead increases with the increasing number of layers, featuring a long-range percolative charge transport through the 3D network connectivity. Related tests on the electrical conductivity of Pt nanomeshes, obtained via SS-LZA over macroscopic areas and LPI, were performed by Majewski et al. [33]. In addition to confirming the high resistivity of a single layer of Pt nanowires, interestingly they found both electrical and optical anisotropy of the nanowire arrays (Figure 11b). On the contrary, the electrical conductivity of the two-layer Pt nanomeshes is nearly isotropic when measured along and across the nanowire axis, thus highlighting the determining role of the nanoscale patterning in tuning the functional properties.

Very recently, Jin and coworkers [53] have stated the compatibility of the laser writing process and both LPI and conventional semiconductor processes, thus paving the way to applications involving also fab-friendly processes. Indeed, they realised gold nanowires through the Au ionic loading into high-χ PS-*b*-P2VP previously aligned via laser writing performed on common untreated silicon substrates. Other notable results include those already mentioned by Shin and coworkers [27], who realised mixed Au nanowires and Pt nanodot morphologies by directed BCP self-assembly. The authors demonstrated yet another application for these complex structures, testing them for the creation of charge trap memory devices as well (Figure 11c). They addressed the bottleneck due to charge leakage, which has been limiting the further downscaling of the memory cell dimension so far. The possibility of employing a variety of multicomponent combinations allows the precise control of the memory window of the device, ranging from the 2.2 V window of the Au nanodot device, through the 3.0 V window of the Pt nanodot array, to the 3.6 V memory window of the Pt-Au binary nanodot device, as shown by the capacitance-voltage responses. In the latter case, the combination of multimetallic elements allows the tunability of charge trap characteristics, further stabilising the memory operation.

Another, rather exotic, example of LPI of interest for electronic applications concerns the realisation of thin perovskite nanopattern already mentioned earlier [21]. The LPI in BCPs nanodomains allows the challenge of its downsizing to the sub-100 nm scale to be overcome, due to the ionic characteristics which make halide perovskites (HPs) highly unstable in ambient conditions. HPs appear to be effectively stabilised via their well-controlled crystallisation directly into BCP self-assembled templates. The possibility of nanopatterning thin perovskite films without harming their excellent photo-electronic properties opens up their suitability for a wide range of optoelectronics applications, including diodes, light-emitting nonvolatile memories, lasing, and metasurfaces. In more detail, Han and coworkers were able to successfully employ their hierarchically ordered MAPbBr_3_/PS-*b*-P2VP film as a photo-response layer in a two-terminal parallel-type photodetector, thanks to the enhanced photoconduction of the so-synthesised HPs. Indeed, the MAPbBr_3_/P2VP matrix efficiently served as a photocurrent pathway, although with anisotropic features due to the nanoimprinting directionality. It is worth noting that these devices showed both monotonically increasing photoresponsivity with increasing light intensity and notable photocurrent switching properties under an input ON/OFF illumination.

### 5.3. Other Applications

The interest in the use of self-assembled BCP for the nanopatterning of networked structures has been extended by the perspective of a wide range of applications, spanning catalysis to the fabrication of functional advanced materials and biocompatible substrates. Metallic structures, such as the platinum nanomeshes fabricated by Subramanian et al. [17], have been reported as promising active media for catalytic processes and fuel cell electrodes since the control of their morphology and dimension can respond to the need for tuning the electrocatalytic activity of nanostructured materials. Indeed, so-synthesised platinum nanowires could improve Pt performance in fuel cells, thus representing a crucial step forward in the design of efficient direct formic acid fuel cells (DFAFC), alongside similar results already achieved via other synthesis methods [54]. Some catalytic applications have also been envisioned in nanoporous ceramics, fabricated through hybrid methods involving metal infiltration from the liquid phase [55]. The formation of palladium centres directly embedded in a nanoporous alumina matrix resulted in PdO nanoparticles that were successfully tested as catalysts with high-temperature stability for both carbon monoxide oxidation and methane combustion reactions. Other diverse impressive results come from BCMN, where the ability of the soluble polymer block to crystallise and the micellisation of the copolymer in solution is deemed crucial to control the properties of the resulting networks [56]. BCMN through Au nanoparticles has been demonstrated to enable the low-temperature growth of Si 1D materials on borosilicate glass (BSG) and SiOx/Si substrates with fine control over the size and spacing of Si nanowires and nanotubes [57]. Glass and coworkers [30,50] have largely investigated the combination of BCMN and LPI with other conventional lithographic methods. Nanodot size and spacing are controlled through the amount of the metal loaded and the molecular weight, respectively, while the separation among groups of nanoparticles is regulated by photo or e-beam lithography pre-patterning steps. With this method, a few nanometre-sized Au nanoclusters have served to design uniformly structured interfaces onto solid supports.

Glass and colleagues also reported the use of monomicellar films as negative resist for EBL, thus overcoming the inherent limitations with standard resist materials. The outstanding result achieved using metals and metal oxides as cluster materials allows the extension of this lithographic method to non-conductive supports such as common glass cover slips in addition to common conductive substrates [50]. The enormous mechanical stability demonstrated by these nanopatterns on the substrates makes them available for biological applications such as templates for immobilising single proteins and probing their interactions with living cells. Promising biological screening applications of BCMN and LPI have been reported by Lohmüller et al. [31]. The authors obtained nanoparticle-patterned substrates combining BCMN with standard lithographic methods, thus increasing the processing rate and circumventing the limitation of BCMN to inorganic supports (glass or silicon). They succeeded in transferring the nanoparticles on polymeric substrates, with great perspective for applications in cell biology and tissue engineering.

## 6. Conclusions and Perspectives

The countless possibilities offered by LPI, described in large part in this review, clearly show an enormous application potential. The realisation of metamaterials [58], meso-porous superconductors [59], artificial cell membrane arrays [60], SERS substrates [49] and physically unclonable functions (PUF) [61] based on metallic nanostructures are just a few examples of how LPI of BCPs could pave the way for new discoveries in the field of fundamental science. For some of the application areas described, however, extreme control over the intrinsic properties of the nanostructures (e.g., electrical resistivity and refractive index) is required, while maintaining unaltered geometric structures at the nanoscale. These properties strongly depend on the size and shape of the nanostructures and the chemical state of the metal infiltrated inside them. To control and optimise these characteristics, it is necessary to develop hybrid metrology protocols, able to reliably put together measurements operated with very different techniques (e.g., scanning probe and electron microscopies and X-ray scattering techniques). This is the most complex challenge that must be quickly overcome to fully exploit the capabilities of the liquid phase infiltration method.

## Figures and Tables

**Figure 1 polymers-14-04317-f001:**
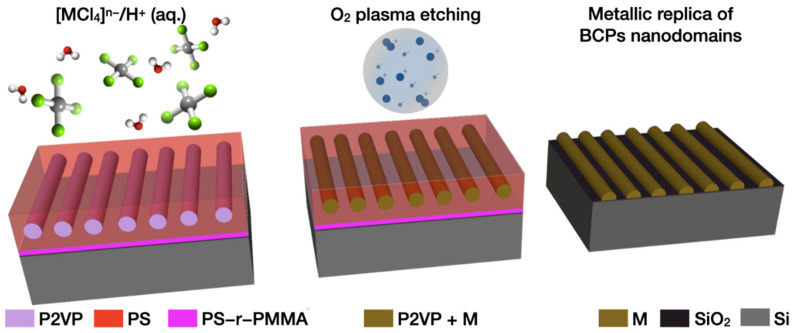
Schematics of liquid phase infiltration of BCPs. Horizontal cylindrical PS-*b*-P2VP are immersed in an acidic solution of the metal salt precursor [MCl_4_]^−^. The anionic metal complexes selectively interact via electrostatic interaction with the protonated P2VP. The subsequent brief exposure to O_2_ plasma selectively removes the polymer and reduces the metal salt precursor located in the P2VP domain into metallic nanostructures, perfectly replicating the BCP template morphology.

**Figure 2 polymers-14-04317-f002:**
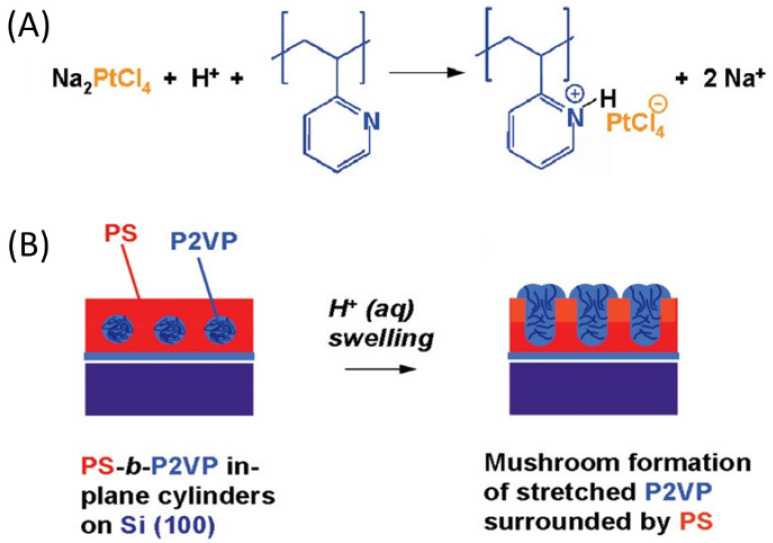
(**A**) Mechanism for the selective metal salt precursor interaction with P2VP in acidic conditions. (**B**) Acid-induced surface reconstruction. The repulsive interactions among the protonated pyridine groups determine the selective swelling of the P2VP domain, fully exposing the reactive sites to the metal salt solution. Adapted with permission from Ref. [15]. Copyright (2008) American Chemical Society.

**Figure 3 polymers-14-04317-f003:**
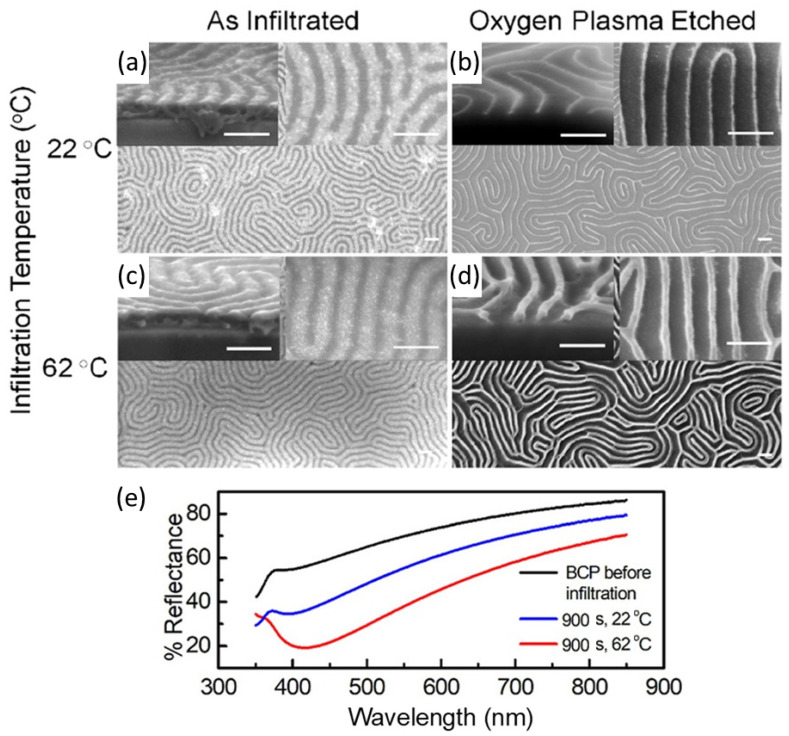
Top-view and cross-sectional SEM images of self-assembled PS-*b*-P2VP infiltrated with Pt at 22 °C and 62 °C before (**a**,**b**) and after (**c**,**d**) the selective removal of the polymeric template. Scale bars set at 100 nm. (**e**) Spectroscopic optical reflectance spectra of PS-*b*-P2VP infiltrated at 22 °C and 62 °C. Adapted with permission from Ref. [17]. Copyright (2020) American Chemical Society.

**Figure 4 polymers-14-04317-f004:**
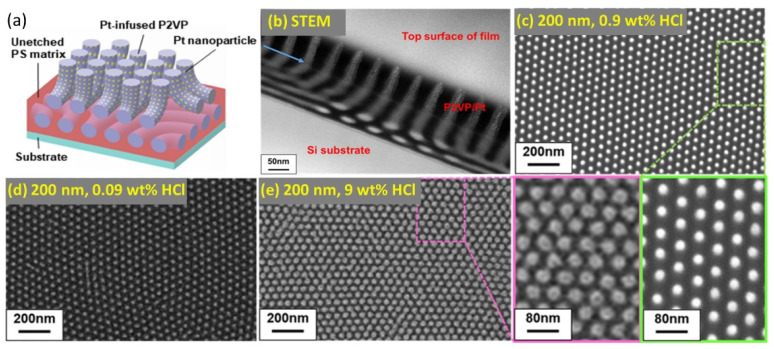
(**a**) Schematic illustration of Pt-infused multilevel PS-*b*-P2VP and (**b**) relative cross-sectional STEM image. Dimensional evolution of a 200 nm thick BCP film, for (**c**) 0.9, (**d**) 0.09 and (**e**) 9 wt % HCl concentrations in metal precursor solutions. Higher magnification SEM images on insets of (**c**,**e**). Adapted with permission from Ref. [23]. Copyright (2021) American Chemical Society.

**Figure 5 polymers-14-04317-f005:**
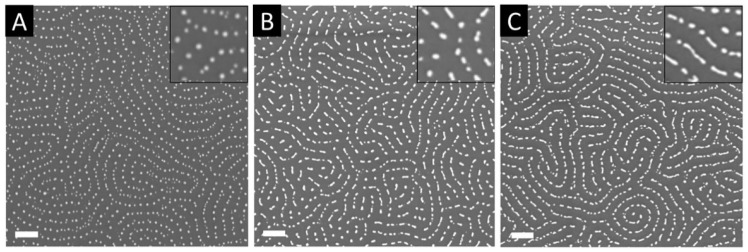
SEM images of Au nanostructures obtained after immersion of PS-*b*-P2VP into aqueous salt precursor solution for (**A**) 1 h, (**B**) 48 h, (**C**) 120 h and subsequent O_2_ plasma reduction (scale bars set at 100 nm). Reproduced from Ref. [32] with permission from the Royal Society of Chemistry.

**Figure 6 polymers-14-04317-f006:**
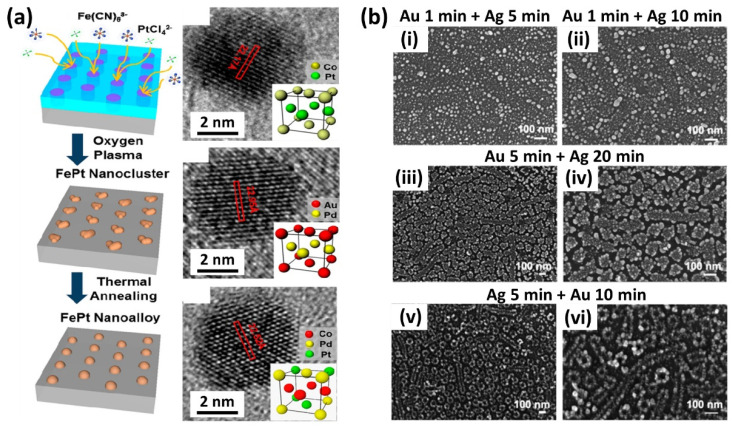
(**a**) Schematics of the general route for the simultaneous infiltration of multiple metal salt precursors and relative nanoalloys (CoPt, AuPd, CoPdPt) obtained after selective polymer removal. Adapted with permission from Ref. [24]. Copyright (2013) American Chemical Society. (**b**) SEM micrographs of AuAg nanostructures obtained by LPI of 0.1 mM HAuCl_4_ and 0.1 mM AgNO_3_ in 0.9% HF for different infiltration timings. (**i**) Infiltration of Au for 1 min followed by infiltration of Ag for 5 min. (**ii**) Infiltration of Au for 1 min followed by infiltration of Ag for 10 min. (**iii**,**iv**) Infiltration of Au for 5 min followed by infiltration of Ag for 20 min at different magnifications, respectively. (**v**,**vi**) Infiltration of Ag for 5 min followed by infiltration of Au for 10 min at different magnifications, respectively. Adapted with permission from Ref. [35]. Copyright (2006) American Chemical Society.

**Figure 7 polymers-14-04317-f007:**
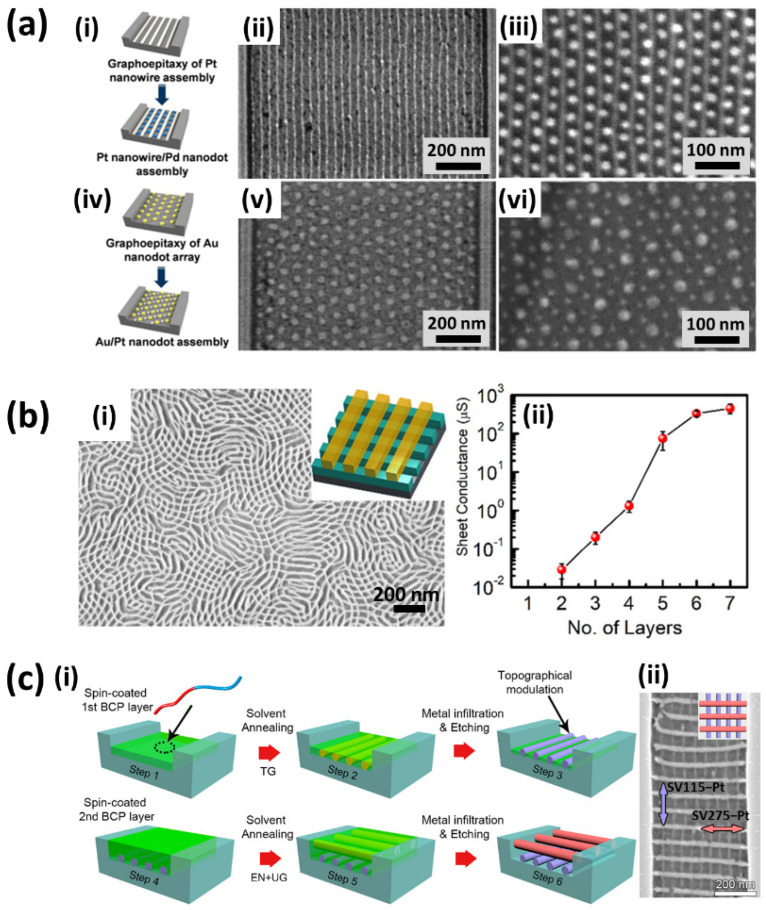
Complex structures obtained by implementing liquid phase infiltration with (**a**) DSA. (**i**,**iv**) Represent the schematic procedure of DSA by two step process and SEM micrographs of laterally ordered (**ii**) Pt nanowires, (**iii**) Pt nanowire with Pd nanodots, (**v**) Au nanodots and (**vi**) Au−Pt nanodots. (**b**) Sequential multilayer self-assembly. (**i**) Illustrates a two−layer Pt nanomesh and (**ii**) the layer−dependent electrical conductance of Pt nanomesh. (**c**) MMDSA. (**i**) Illustrates the schematized major steps of MMDSA process and (**ii**) the resulting Pt nanomesh. (**a**) Adapted with permission from Ref. [27]. Copyright (2013) American Chemical Society. (**b**) Adapted with permission from Ref. [17]. Copyright (2020) American Chemical Society. (**c**) Adapted with permission from Ref. [43]. Copyright (2021) American Chemical Society.

**Figure 8 polymers-14-04317-f008:**
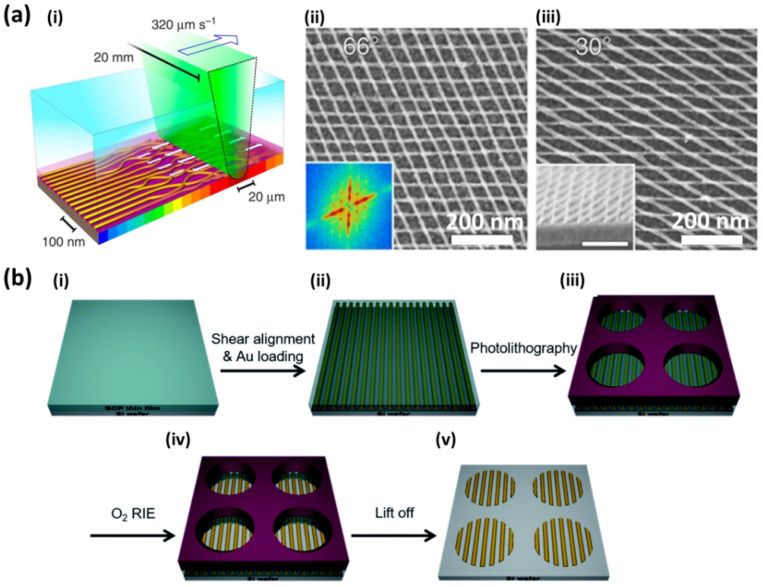
Complex structures obtained by implementing liquid phase infiltration with (**a**) SS-LZA. (**i**) Schematically represents the experimental setup and (**ii**,**iii**) the SEM micrographs of double−layered Pt nanowires with different hatching angles of 66° and 33°, respectively. (**b**) Shear-induced self-assembly and photolithography schematics. (**i**,**ii**) The shear is applied to the deposited BCP thin film followed by infiltration of Au. (**iii**) Micropatterns are then created by photolithography. Subsequent (**iv**) O_2_ RIE and (**v**) lift-off process reveal aligned Au nanopatterns. (**a**) Adapted under the terms of Creative Commons Attribution 4.0 Licence from Ref. [33]. Copyright 2015, the authors, published by Springer Nature. (**b**) Reproduced from Ref. [44] with permission from the Royal Society of Chemistry.

**Figure 9 polymers-14-04317-f009:**
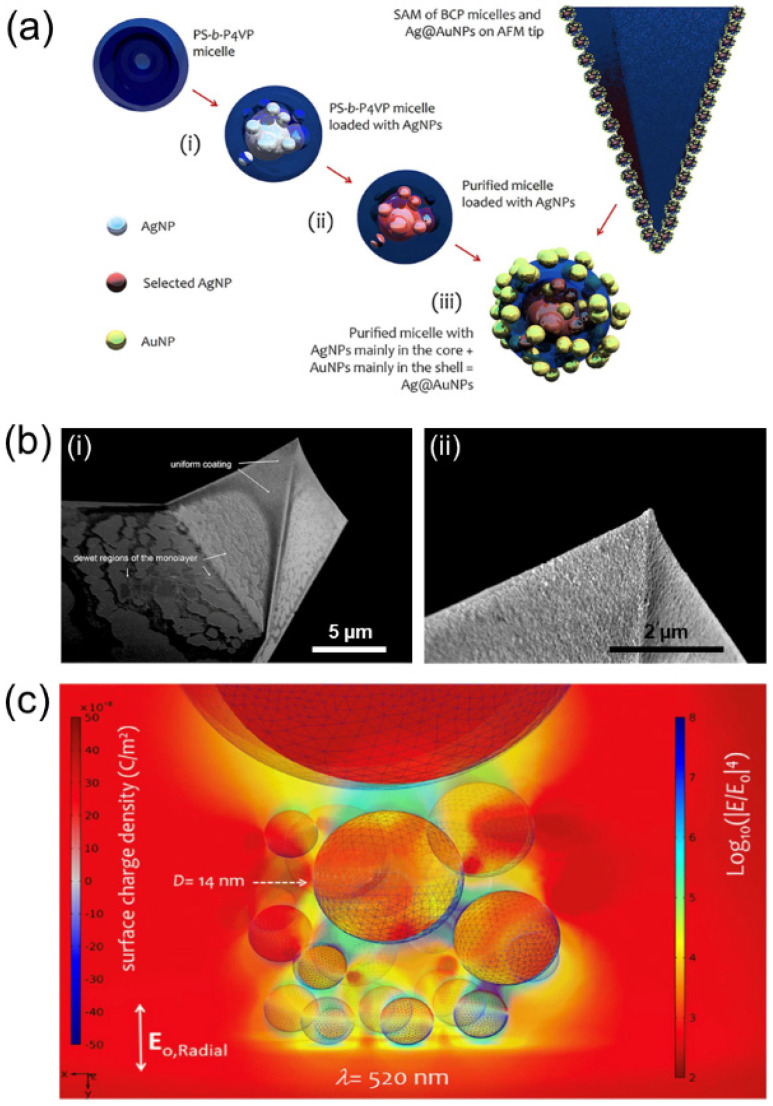
(**a**) Schematics of the bi-metallic LPI process in PS-*b*-P4VP micelles and coating of the AFM tip. (**b**) SEM images of the AFM tip at different magnifications (**i**,**ii**), showing the formation of a uniform layer at the tip apex and a sparsely covered region at the base. (**c**) Numerical simulation of the spatial distribution of the electric field enhancement factor at the tip apex. The scattered electric field E is normalised to the incoming field E_0_ at 520 nm from bottom illumination and surrounding medium enhancement factor *n* = 1.4. Reproduced from Ref. [48] with permission from Springer Nature.

**Figure 10 polymers-14-04317-f010:**
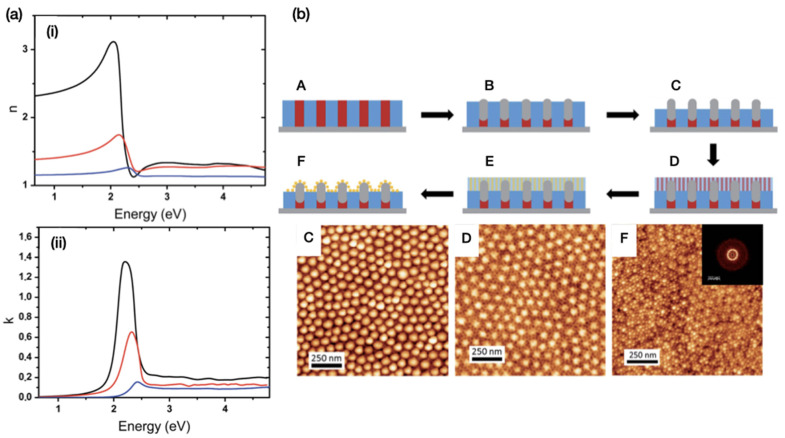
(**a**) Two graphs reporting the optical constants (**i**) *n* and (**ii**) *k* for the samples treated with 1 h (blue line), 48 h (red line) and 120 h (black line) of immersion time in the aqueous gold precursor solution for the LPI whose SEM images are already reported in Figure 5. Reproduced from Ref. [32] with permission from the Royal Society of Chemistry. (**b**) Fabrication scheme for the realisation of hybrid raspberry-like clusters including one self-assembled BCP layer (**A**) infiltrated with Al_2_O_3_ via SIS (**B**), preparation and self-assembly of a second BCP layer with different periodicity (**C**–**D**), LPI of Au (**E**) and metal reduction forming decorative nanoparticles (**F**). The AFM micrographs below correspond to the previous steps (**C**), (**D**) and (**F**). The inset reports the FFT highlighting the second iterative self-assembly at lower periodicity already visible in the AFM images. Reproduced from Ref. [42] with permission from the Royal Society of Chemistry.

**Figure 11 polymers-14-04317-f011:**
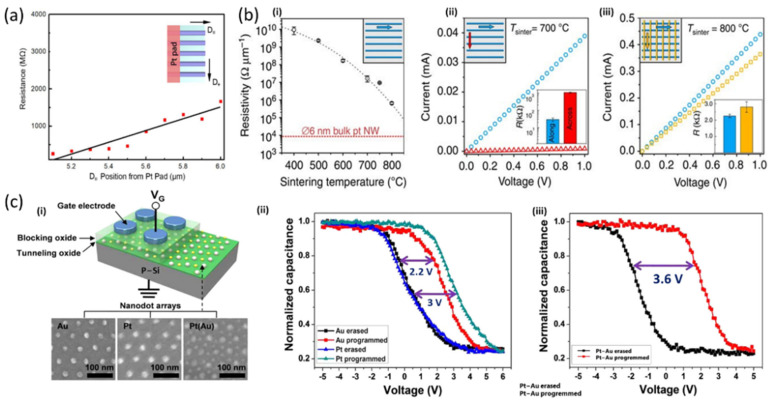
(**a**) Resistance of Pt nanowires measured with conductive atomic force microscopy at a varying distance away from the Pt pad parallel to the wires. Adapted with permission from Ref. [43]. Copyright (2021) American Chemical Society. (**b**) (**i**) Electrical resistivity measured at room temperature as a function of sintering temperature. I–V characteristics of (**ii**) the single-layered and (**iii**) the two-layer Pt nanowire arrays, measured along two orthogonal directions. Adapted under the terms of Creative Commons Attribution 4.0 Licence from Ref. [33]. Copyright 2015, the authors, published by Springer Nature. (**c**) (**i**) Schematic illustration of the charge trap memory device architecture. Capacitance-voltage responses of the devices with (**ii**) Pt or Au nanodot array and (**iii**) Pt-Au binary nanodot array, upon programming/erasing operation at 1 MHz. Adapted with permission from Ref. [27]. Copyright (2013) American Chemical Society.

## Data Availability

The data presented in this study are available on request from the corresponding author.

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
