# Peer review of "Liquid Phase Infiltration of Block Copolymers"

_polymers, 2022, doi:10.3390/polym14204317_

Round 1

Reviewer 1 Report

The presented review nicely shows the review of the process of Liquid Phase Infiltration (LPI) of Block Copolymers (BCPs), leading to the synthesis of inorganic materials. The review provides an overview of the mechanism involved in the LPI, outlining the role of the different polymer infiltration parameters on the resulting material properties. In the end, the application of the LPI of BCPs in photonics, plasmonics and electronics, as well as applications in different domains of materials, were described. The literature was chosen sufficiently. Only some minor English check is required.

Author Response

Dear Editor,

We are returning the manuscript polymers-1959510 after revision according to the Reviewer’s comments.

First of all we would like to thank Reviewers for the positive comments to our paper. According to their suggestions we improved the quality of the figures, we reviewed the English language and style, and we removed the acronyms from the title. 

Attached to this email you cand find the corrected version of the manuscript in which all changes have been underlined. 

Bests regards

Federico Ferrarese Lupi

Reviewer 2 Report

Comments to the authors: This manuscript provides an overview of the mechanism involved in the liquid phase infiltration (LPI), including newly developed methodologies that extend the LPI to the realization of multicomponent and 3D inorganic nanostructures. Also, the recently reported implementation of LPI into different applications such as photonics, plasmonics and electronics are highlighted. I found the manuscript to be very well written and addressed the important aspects of LPI of block copolymers. I would recommend the manuscript to be considered for publication after addressing the following concerns.

Comment:1

The title should not contain acronyms unless the acronym is exclusively known or widely used by its acronyms.

Comment:2

Figures have very poor image quality, especially figures 6, 7, and 8. In those figures, the text needs to be more clearly visible.

Comment:3

The scale bar is not visible in all SEM images. Need to mention all the scale bars clearly.

Author Response

(The authors gave the same response as above.)
